# End-to-End Learning for Visual Navigation of Forest Environments

Chaoyue Niu *, Klaus-Peter Zauner and Danesh Tarapore

School of Electronics and Computer Science, University of Southampton, Southampton SO17 1BJ, UK
* Correspondence: cn1n18@soton.ac.uk

**Abstract:** Off-road navigation in forest environments is a challenging problem in field robotics. Rovers are required to infer their traversability over a priori unknown and dynamically changing forest terrain using noisy onboard navigation sensors. The problem is compounded for small-sized rovers, such as that of a swarm. Their size-proportional low-viewpoint affords them a restricted view for navigation, which may be partially occluded by forest vegetation. Hand-crafted features, typically employed for terrain traversability analysis, are often brittle and may fail to discriminate obstacles in varying lighting and weather conditions. We design a low-cost navigation system tailored for small-sized forest rovers using self-learned features. The MobileNet-V1 and MobileNet-V2 models, trained following an end-to-end learning approach, are deployed to steer a mobile platform, with a human-in-the-loop, towards traversable paths while avoiding obstacles. Receiving a 128 × 96 pixel RGB image from a monocular camera as input, the algorithm running on a Raspberry Pi 4, exhibited robustness to motion blur, low lighting, shadows and high-contrast lighting conditions. It was able to successfully navigate a total of over 3 km of real-world forest terrain comprising shrubs, dense bushes, tall grass, fallen branches, fallen tree trunks, and standing trees, in over five different weather conditions and four different times of day.

**Keywords:** off-road visual navigation; end-to-end learning; multiclass classification; low-viewpoint forest navigation; low-cost sensors; small-sized rovers; sparse swarms

## 1. Introduction

An estimated 3 trillion trees, mostly in forests that cover 30% of the Earth's landmass, are important for maintaining our ecosystems and counteracting climate change [1,2]. The management, maintenance and conservation of forests are enormous operations. Forests need to be adapted to stay resilient in the face of new rainfall patterns, increased wind, more generations of insect pests per year, and the arrival of new pathogens [3]. At present, forests are monitored on a large scale from space [4], and more locally with aerial surveys [5]. However, many aspects of tree growth and health can best be determined from below the canopy, or require access to the ground. Conceivably, a sparse swarm of rovers could assist in monitoring forests [6]. The swarm could gather spatio-temporal information, such as census data on healthy tree saplings, or visually inspect bark and leaves for symptoms of devastating invasive diseases [7]. A swarm could collaboratively estimate the locations of forest areas that are prone to wildfires, enabling precise preventive measures [8]. Importantly, the individual rovers of the swarm have to be small-sized (portable) to reduce their environmental impact, such as from soil compaction [9]. The rovers also have to be inexpensive to allow their large-scale deployment as a swarm.

Off-trail navigation in forest environments is an open problem in field robotics [10]. Forest environments comprise a variety of different vegetation such as leaves, twigs, fallen branches, grass, shrubs, standing and fallen trees, and overhanging bushes. Rovers are required to predict the terrain traversability over a priori unknown forest terrain relying solely on onboard sensors, and do so under varying lighting and weather conditions [11,12]. Furthermore, the prediction of the rover–terrain interactions is not only impacted by terrain and the weather conditions (such as wet versus dry foliage), but also susceptible

to changes experienced by the rover in prolonged operation (e.g., mud sticking to the rover's wheels) [13]. For small rovers with a low camera viewpoint, which is easily occluded by compliant vegetation such as grass or overhanging leaves, forest navigation is especially challenging.

Off-road terrain traversability for ground robots has been investigated in numerous studies (c.f. [14,15]), often motivated by the DARPA programs [16,17]. Machine learning algorithms for off-road navigation typically utilize hand-crafted features [18] engineered by experts based on the application scenario and the rover's operating environment. In structured environments, these rely mainly on geometry (e.g., slope, step and roughness features of city walkways [19]) and appearance (color and texture of obstacles [20]). Unstructured environments typically require engineered features of proprioceptive information such as drive electrical currents, acceleration forces and chassis orientation on uneven terrain [18,21], in addition to geometry and appearance-based features. For example, features engineered from proprioceptive sensors, particularly the mean slope of terrain profiles from chassis orientation, were used in [22] for mobility prediction models.

Hand-crafted features have limitations for terrain traversability analysis in off-road environments. Features engineered from geometric data, such as terrain roughness and slope, are often unreliable in unstructured environments due to limited depth information [23,24]. Estimated digital elevation maps can be incomplete due to occlusions [23]. Compliant vegetation, such as high grass, is difficult to be captured with engineered geometry-based features [24]. Hand-crafted visual features (e.g., color and textural descriptors) suffer from environmental factors such as high-contrast lighting [25,26]. In summary, hand-crafted features are impaired by engineering bias and often lead to poor discriminative power [27]. Hand-crafted features that are robust to compliant objects, deep shadows, and motion blur are complicated to engineer, computationally expensive to run, and often brittle in varying environmental conditions.

In contrast to hand-crafting, many recent studies have turned to self-learned features trained using end-to-end learning to directly output steering actions for a rover (see Figure 1 and Table A1). Steering prediction algorithms following end-to-end learning have been successfully applied in structured environments, such as in mazes [28,29], following colored tracks [30–33] and corridors [25,34–36]. In outdoor environments, algorithms using end-to-end learning have demonstrated some promising results in autonomous driving on well-paved roads in structured urban environments under varying lighting and weather conditions [37–40]. Moreover, recently, a few studies have investigated end-to-end learning algorithms for off-road navigation [41–43]. For example, using control policy predictions of steering and throttle commands trained using an end-to-end imitation learning approach, a 1/5-scaled RC vehicle was able to successfully navigate a dirt track—without obstacles—at high speed [41]. However, the application of end-to-end learning for small-sized rovers navigating forest environments comprising a variety of compliant (grass, shrubs) and rigid obstacles (fallen branches, tree stems) from a viewpoint tens of centimeters off the ground remains to be investigated.

We propose a low-viewpoint navigation system for small-sized forest rovers, trained using end-to-end learning. This approach targets the uncharted bottom-right region in Figure 1. A mobile platform is designed to easily capture and automatically label training data of forest scenes, RGB images at a low-viewpoint. Four different state-of-the-art lightweight convolution neural networks—DenseNet-121, MobileNet-V1, MobileNet-V2 and NASNetMobile—have been investigated for multiclass classification of steering actions. The models are trained using real-world forest data captured from the Southampton Common woodlands (Hampshire, UK). From the four models, the MobileNet-V1 and MobileNet-V2 are selected for field experiments due to their high accuracy and runtime performance. To sidestep the additional challenges of designing a high-endurance locomotion system for a small-sized low-cost rover, in this study, we focus solely on the navigation system. Therefore, the mobile platform is pushed manually by an operator, guided by the steering actions of the classification model running on a Raspberry Pi onboard the platform.

The developed low-viewpoint navigation algorithm uses a 128 × 96 resolution RGB image. Navigation using the developed classification model has been extensively tested in field trials, successfully navigating a total of over 3 km of real-world forest terrain under five different weather conditions and four different times of day, including high-contrast sunlight and low-lighting at dusk.

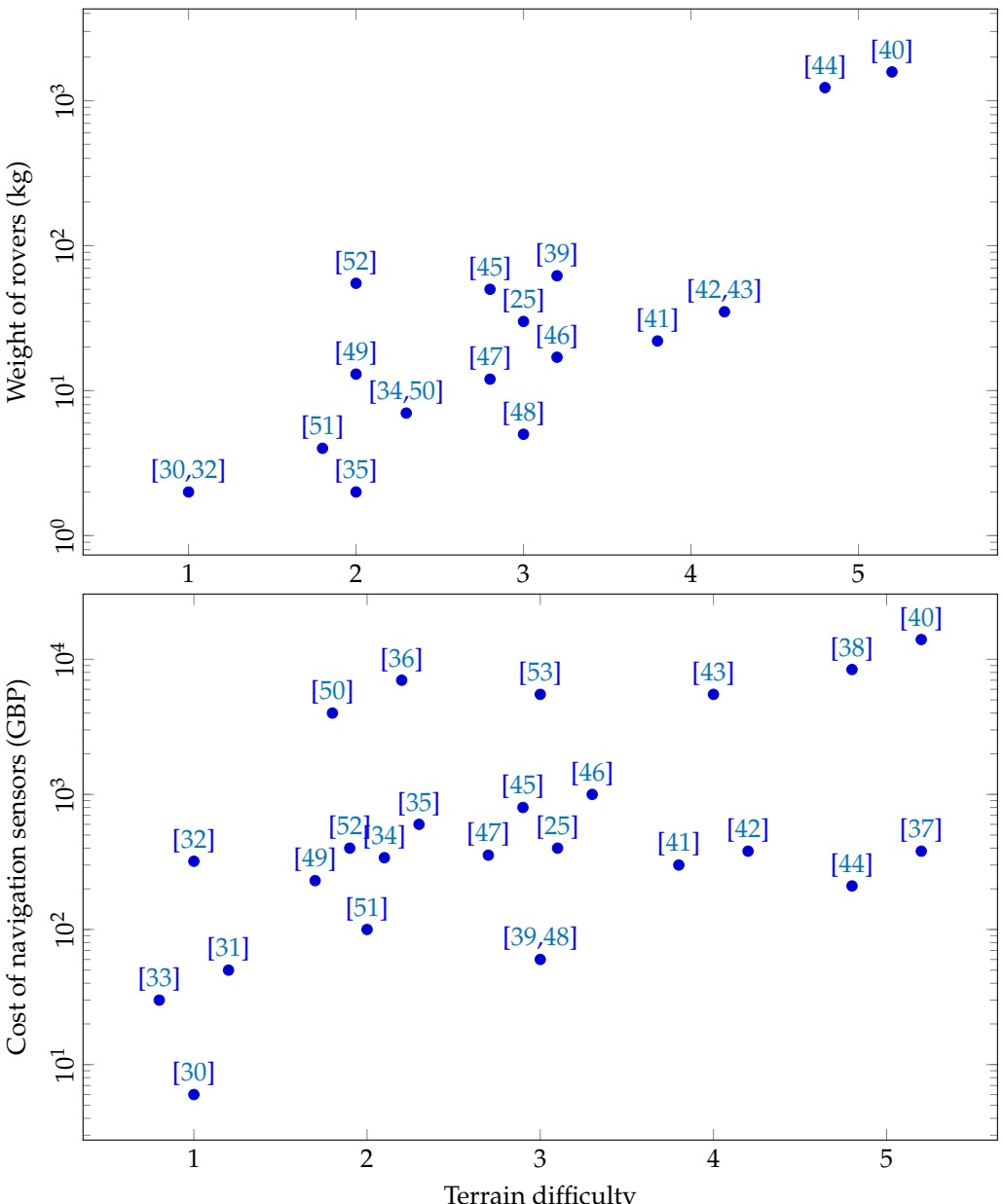

**Figure 1.** The relationship between the navigation sensor cost, rover weight and terrain difficulty in studies using end-to-end learning. Terrains are categorized in ascending order of difficulty as follows: 1. colored track indoors; 2. corridor and rooms indoors; 3. sidewalks and walkways in urban environment; 4. off-road on cemented paths, short grass, pebbles, dirt and dry leaves; and 5. highways and traffic roads, forest environment—dense bushes, tall grass, fallen branches, fallen tree trunks, standing trees, small mounds and ditches. Importantly, to the best of our knowledge, none of the studies have investigated end-to-end learning for navigation in forest environments. For details on environments in the referenced studies, see Table A1 in Appendix A [30–53].

## 2. Materials and Methods

Our algorithm for forest navigation employs deep neural networks to train a multiclass classification model following end-to-end learning. Training data for the classification comprise RGB images and corresponding steering actions, obtained by an operator pushing the mobile platform through the forest. The trained models infer steering actions from RGB images in real-time, on an onboard embedded computer, with sufficient accuracy to facilitate navigation of the forest environment.

Training data for forest navigation: Data was collected at the Southampton Common (Hampshire, UK), a large area of over 1.48 km$^2$ featuring woodlands, rough grassland, ponds, wetlands and lakes. Several paths of the woodlands, both on-trail and off-trail and totaling over 600 m, were selected for recording data. The selected paths comprised a number of different obstacles such as grass, bushes, fallen tree branches, leaf litter, and fallen and standing trees.

The customized mobile platform was manually pushed by an operator along the paths to be recorded (for platform details, see [11]). On the platform, two incremental photoelectric rotary encoders were attached to a CamdenBoss X8 series enclosure box (L × W × H: 18.5 × 13.5 × 10 cm). Two black polyurethane scooter wheels were mounted on either side of the enclosure, one for each encoder. The wheels were 10 cm in diameter and 2.4 cm in width to enable traversal over rough terrain. The encoders were connected to a Micropython enabled Adafruit ItsyBitsy M4 Express ARM board, which made the time stamped rotary encoder readings available over a USB connection. The enclosure was mounted at the end of a 1.21 m telescopic extension pole, allowing the operator to roll the enclosure on its wheels along the ground by pushing it forward while walking. Inside the enclosure, an Intel RealSense D435i camera was mounted 15 cm above the ground with a free field of view in the direction of motion. The rotary encoder data were time synchronized with the recorded RGB images from the camera at 30 frames per second, and recorded at the same rate. A laptop computer connected to the camera, and to the USB connection from the rotary encoders, was used to store the data.

The operator pushed the mobile platform along forest paths while performing go straight (GS), turn left (TL), turn right (TR) and go back (GB) actions. All the actions were performed as discrete movements to ensure the wheels of the mobile platform rotated smoothly on challenging forest terrain, to provide reliable encoder data. With the GS action, the platform was pushed straight approximately 50 cm forward. Rotary actions of TL and TR pivoted the platform by approximately 15° along the yaw axis. Finally, the GB action rotated the platform by approximately 180°. The actions allowed the operator to navigate the mobile platform through the forest, avoiding collisions by steering around the obstacles, and turning around when there were no traversable paths to circumvent the obstacles.

In total, 29,005 RGB images were recorded by the mobile platform. To automatically label the recorded RGB images for training the multiclass classification models, the left and right wheel encoder data were used to label the corresponding timestamped RGB images. The images were labeled as one of GS, TL, TR and GB according to the steering angle of the platform. A few of the GB labeled RGB images had to be manually re-labeled as TL or TR (around 1% of the recorded data), if there was a traversable path on the far left or far right of the image, respectively. Following the labeling, we had 19,573, 3037, 3527, and 2868 images for the GS, TL, TR and GB actions, respectively. Subsets of the recorded data were used for training (70%), validation (15%) and testing (15%) the multiclass classifier models (see Table A2 in Appendix A for details).

**Classification models:** The multiclass classification models are required to infer steering directions (GS, TL, TR and GB) from input RGB images in real time. As the models are to be deployed on low-cost embedded computers, for our study, we compare four state-of-the-art light-weight neural networks—DenseNet-121 [54], MobileNet-V1 [55], MobileNet-V2 [56] and NasNetMobile [57]. The implementations of these networks are available at Keras (Keras is a deep learning API written in Python, see https://keras.io/api/applications, accessed 23 November 2020)

The initial weights of the DenseNet-121, MobileNet-V1, MobileNet-V2 and NasNet-Mobile models had been pre-trained on the ImageNet dataset [58] to speed up the model convergence. Subsequently, we unfroze all the layers in the investigated models and re-trained them on our forest data. For steering direction prediction, a flattened convolutional layer followed by three fully connected (FC) layers were added to the models (see the architecture in Figure 2). The first two FC layers had a rectified linear unit activation, and the last FC layer employed a softmax activation for steering direction selection. Batch normalization and dropout operations (probability of 0.2) were employed after each FC layer to prevent overfitting of the data [59]. For a fair comparison across the models, all the RGB images in the training data were downsampled to 224 × 224. Each of the models were trained for 20 epochs with a batch size of 16, using the Adam optimizer with a categorical cross-entropy loss (log-loss) function [60]. All of the models were implemented in TensorFlow [61] and Keras [62], and trained on a NVIDIA GTX 1080ti (11G) GPU. Training took approximately 16h (DenseNet-121), 7h (MobileNet-V1), 8h (MobileNet-V2) and 10h (NASNetMobile) on a NVIDIA GTX 1080ti (11G) GPU for 224 × 224 resolution RGB images. The trained Tensorflow models were compiled into TensorFlow-Lite models [63], resulting in over a ten-fold improvement in runtime performance.

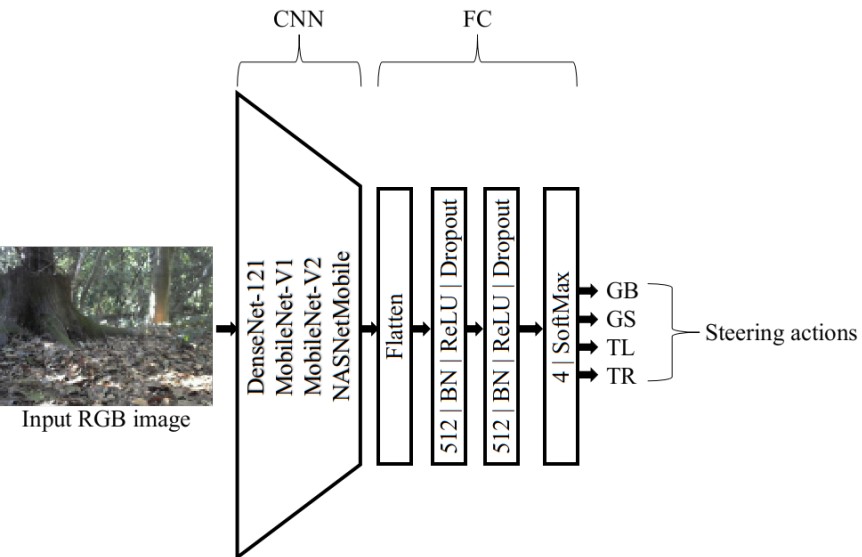

**Figure 2.** Architecture of multiclass classification models for end-to-end learning. The input RGB image is first fed into lightweight convolutional neural networks—one of DenseNet-121, MobileNet, MobileNet-V2 and NASNetMobile—that are pre-trained on ImageNet. Outputs of the convolutional neural networks are flattened and input into three fully connected (FC) layers. The first two layers utilize a rectified linear unit activation (ReLU). A softmax activation is utilized by the final layer for steering direction selection—one of go straight (GS), turn left (TL), turn right (TR), and go back (GB). Batch normalization (BN) and dropout operations were employed after each FC layer to avoid overfitting the training data.

The trained DenseNet-121, MobileNet-V1, MobileNet-V2 and NASNetMobile models all achieved a high classification performance on the tested RGB images (see the accuracy and log-loss in Table 1). In particular, all four models were largely able to accurately classify the GS, TL, TR and TB steering actions, with the DenseNet-121 model attaining a high overall accuracy across all four classes (see confusion matrix in Figure 3). The high accuracy in steering action classification was further supported by a 5-fold cross-validation (see the details in Table A4 in Appendix A). However, the DenseNet-121 model was impaired by a high runtime, requiring over twice the time as the other models to classify the images on a Raspberry Pi 4 (see the runtime in Table 1). Therefore, in considering the tradeoff between accuracy and runtime, the MobileNet-V1 and MobileNet-V2 were selected for field experiments.

For the field experiments, the runtimes of the selected MobileNet-V1 and MobileNet-V2 models were improved by downsampling the resolution of the 224 × 224 RGB images input into the model. Therefore, these two models were retrained following the same experimental setup (20 epochs with a batch size of 16) after downsampling the RGB images of the training data to 128 × 128, 128 × 96, 64 × 64, and 32 × 32, in separate and independent experiments. The results from our parameter tuning experiments indicated a steep drop in accuracy at resolutions below 128 × 96 (see Table A3 in Appendix A for MobileNet-V1; similar trends in accuracy were observed for MobileNet-V2). Consequently, the MobileNet-V1 and MobileNet-V2 models trained with 128 × 96 resolution images (see the performance details in Table 2, the 5-fold cross-validation in Table A5 in Appendix A, and the confusion matrix in Figure 4) were deployed for the field experiments.

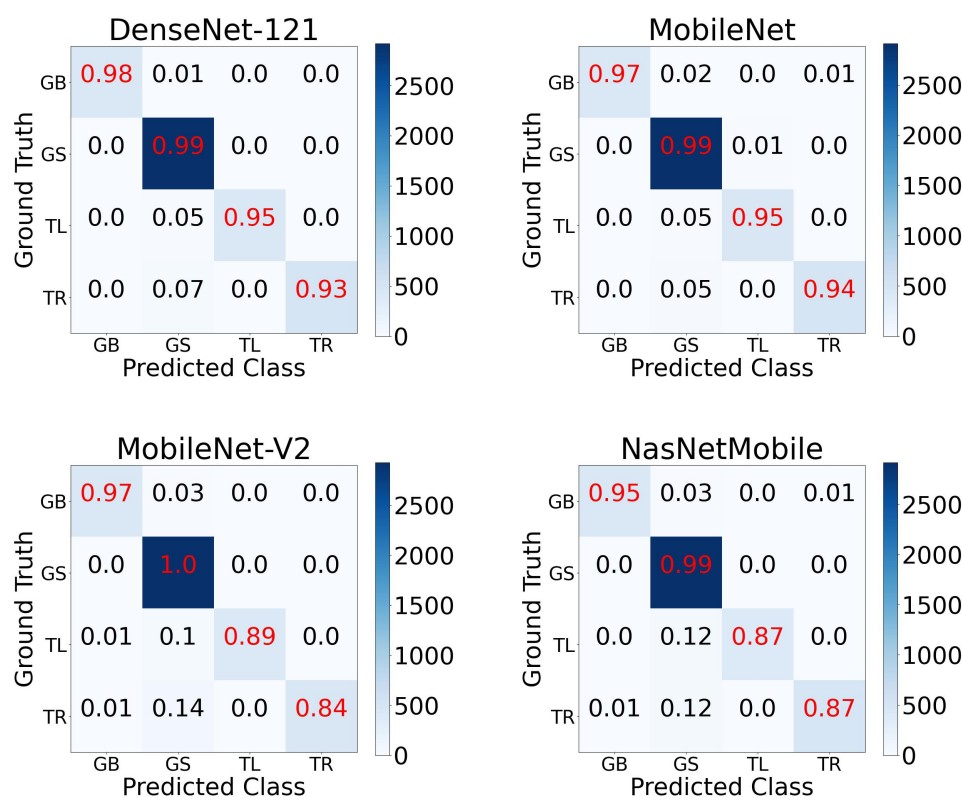

**Figure 3.** Confusion matrix of DenseNet-121, MobileNet-V1, MobileNet-V2 and NASNetMobile multiclass classification models for the go straight (GS), turn left (TL), turn right (TR) and go back (GB) steering actions, with input RGB images of resolutions 224 × 224.

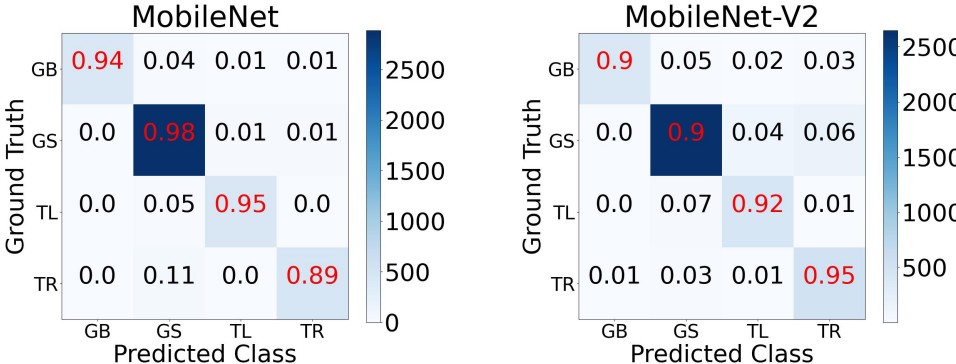

**Figure 4.** Confusion matrix of MobileNet-V1 and MobileNet-V2 models for the go straight (GS), turn left (TL), turn right (TR) and go back (GB) steering actions for input image resolutions of 128 × 96.

**Table 1.** Mean accuracy, log-loss, and mean ± SD runtime of the DenseNet-121, MobileNet-V1, MobileNet-V2 and NASNetMobile classification models for 224 × 224 input RGB images, and the Tensorflow-Lite model size. Accuracy and log-loss were aggregated across 4353 images (testing set). Runtimes were aggregated across 100 randomly selected images, executed on a Raspberry Pi 4.

| Models | Accuracy | Log-Loss | Runtime | Model Size |
|---|---|---|---|---|
| DenseNet-121 | 0.98 | 0.08 | 2.01 ± 0.02 s | 131 MB |
| MobileNet-V1 | 0.98 | 0.12 | 0.78 ± 0.02 s | 116 MB |
| MobileNet-V2 | 0.96 | 0.22 | 0.63 ± 0.01 s | 138 MB |
| NASNetMobile | 0.96 | 0.18 | 1.01 ± 0.01 s | 124 MB |

**Table 2.** Performance of the MobileNet-V1 and MobileNet-V2 models, trained on images of resolution 128 × 96, and selected for field experiments.

| Models | Accuracy | Runtime | Model Size |
|---|---|---|---|
| MobileNet-V1 | 0.96 | 0.43 ± 0.01s | 39 MB |
| MobileNet-V2 | 0.91 | 0.27 ± 0.01s | 41 MB |

**Mobile platform for field experiments:** The mobile platform deployed to assess our multiclass classification models in field experiments was similar to the platform used to gather training data, but with a low-cost RGB webcam for capturing input images, and the addition of display hardware for the output steering commands to be visible to the operator (see the platform and operator in Figure 5). A Logitech C270 HD webcam (diagonal 55° field of view) was mounted inside the enclosure (replacing the Intel RealSense D435i camera) at 18 cm above the ground and was connected to a Raspberry Pi 4. Additionally, a stripboard (9.5 × 12.7 cm) was fixed to two rectangular wooden blocks on the top of the enclosure, alongside two concentric NeoPixel rings of addressable RGB LEDs (Adafruit Industries, New York, NY, USA). The two NeoPixel rings were connected to the Raspberry Pi 4 (4 GB RAM) via a twisted pair (data) and a USB cable (power), while a Schmitt-trigger buffer (74LVC1G17 from Diodes Incorporated, Plano, Texas, USA) in the serial data line was used to overcome the capacitance of the long twisted pair wire. A HERO 9 (GoPro, San Mateo, CA, USA) action camera was also mounted on the telescopic pole 50 cm from the top of the enclosure for a third-person view high-resolution video recording of the field experiments.

The RGB images captured by the Logitech camera every four seconds—one control-cycle—were input into the multiclass classification model deployed on the Raspberry Pi. Subsequently, the classification model output a steering direction—one of GS, TL, TR and GB—which was displayed on the NeoPixel rings (see Figure 5 for details on the direction indications). A fifth action, labeled waypoint, was introduced for the field experiments. The waypoint action superseded the direction outputs of the classification model. It prompted the operator to rotate the platform towards the direction of the goal waypoint. The action occurred every 10 control-cycles and in general could be based on GPS information.

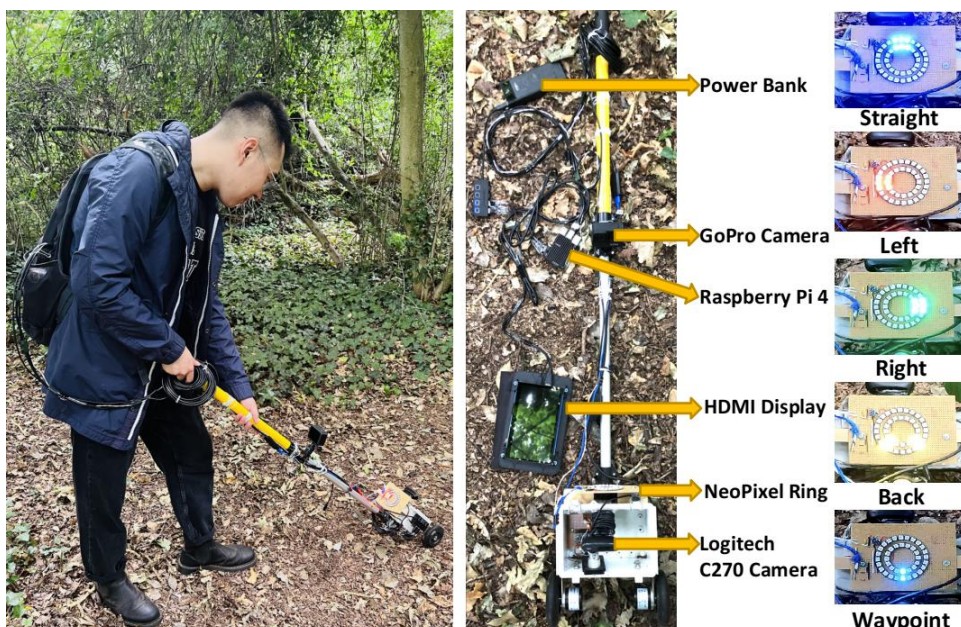

**Figure 5.** The two-wheeled mobile platform with an operator. The platform is equipped with a Logitech C270 camera, two NeoPixel rings, a Raspberry Pi 4, a Raspberry Pi HDMI display, a GoPro camera, and a portable power bank. RGB images captured by the Logitech camera are transmitted to the Raspberry Pi 4, to predict steering directions. The resulting steering actions are displayed on the NeoPixel LED rings. Note that the platform is also used for data collection, where the Logitech camera is replaced with an Intel Realsense D435i camera, and the data including RGB images and rotary encoder counts are synchronously stored on a laptop via a USB connection.

## 3. Experiments

The field experiments to investigate the performance of the developed MobileNet-V1 and MobileNet-V2 multiclass classification models for forest navigation were performed in the Southampton Common woodlands. The experiments were performed for the following two scenarios: (i) following a long forest trail; and (ii) steering through a smaller but more challenging off-trail forest environment.

The performance of the classification models in navigating the forest was assessed with the following metrics: (i) the total distance traversed by the mobile platform to reach its target waypoint; and (ii) turning rate—the proportion of times the mobile platform steered left and right, which is zero for a straight-line trajectory to the target and in general is unbounded (arbitrary long detours and many arbitrary turns without forward progress).

Following a long forest trail: The mobile platform was navigated over a dried mud trail of around 120 m, comprising various compliant and rigid obstacles. Obstacles on and around the trail included dense bushes, tall grass, leaf litter, fallen branches, fallen tree trunks, and standing trees (see examples in Figure 6A).

For the forest-trail experiments, the start and goal waypoints were positioned at (5056.1989 N, 124.0732 W) and (5056.1859 N, 124.1515 W), respectively (see Figure 7). The actions GS, TL, TR, GB and waypoint (defined in Section 2) were used to navigate the mobile platform towards the goal waypoint. As the goal was $210°$ SW of the start location, this bearing was used to rotate the mobile platform to face the goal, using a compass, when the waypoint action was triggered. The GB action was employed by the mobile platform to turn around and attempt to find an alternative path to circumvent large obstacles such as fallen tree trunks. If this action was triggered three times consecutively for the same obstacle, we assumed that there were no traversable paths around the obstacle; consequently, the operator would lift the platform over the obstacle, log the incident, and continue the experiment. The experiment was terminated when the platform reached the goal waypoint.

A

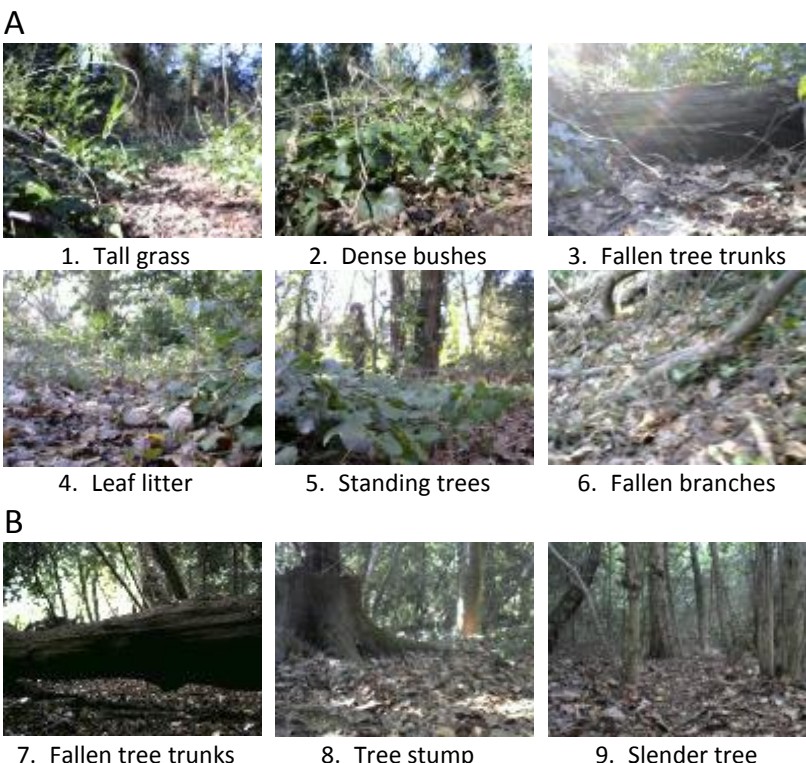

| 1. Tall grass | 2. Dense bushes | 3. Fallen tree trunks |
| 4. Leaf litter | 5. Standing trees | 6. Fallen branches |

B

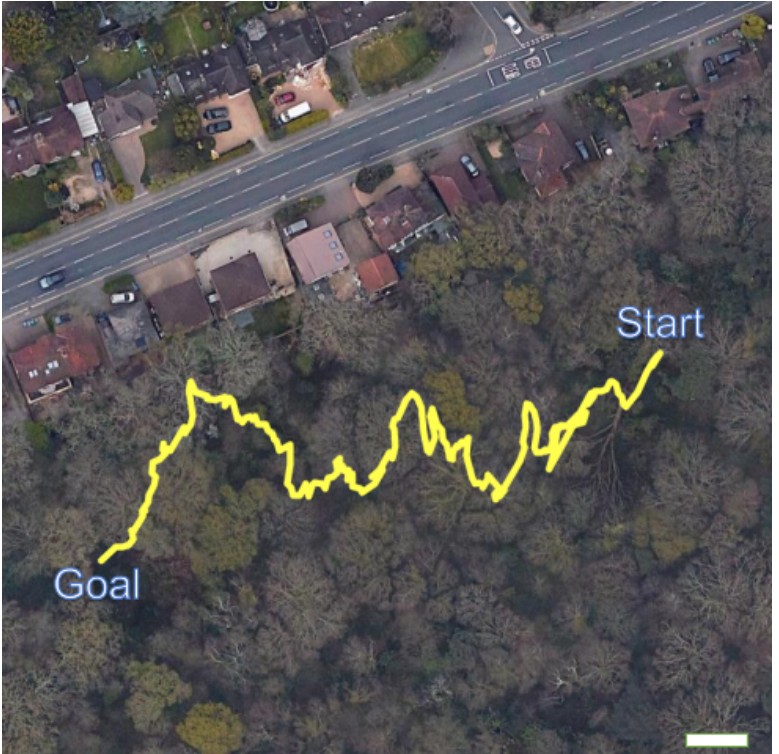

| 7. Fallen tree trunks | 8. Tree stump | 9. Slender tree |

**Figure 6.** Examples of obstacles encountered by the mobile platform both on the forest trail (**A**) and off-trail (**B**) in the Southampton Common woodlands.

**Figure 7.** Trajectory of around 120 m from GPS metadata of the forest trail overlaid on an aerial view of the Southampton Common woodlands. The white scale bar in the lower right corner corresponds to a distance of 10 m. The straight-line distance between the start and goal waypoint is 90 m. Permitted use: Imagery©2022 Getmapping plc, Infoterra Ltd & Bluesky, Maxar Technologies, The GeoInformation Group, Map data©2022 Google.

Forest-trail experiments were performed ten times for each of the MobileNet-V1 and MobileNet-V2 multiclass classification models under several different weather conditions and times of day (see details on the environmental conditions in Table A6 of Appendix A). Across all the experiments, the platform steered by the MobileNet-V1 and MobileNet-V2 models was able to reach the goal waypoint without sustaining any collisions. In navigating with the MobileNet-V1 model, the platform traversed a mean distance of $120 \pm 15$ m with a turning rate of $0.24 \pm 0.02$ (mean $\pm$ SD across ten replicates, see Table 3). While the MobileNet-V2 model was also able to successfully navigate the platform, it was less efficient, steering left and right significantly more often (mean turning rate of $0.52 \pm 0.07$; Kruskal–Wallis test, $p < 0.001$), and accumulating a slightly higher traversed mean distance of $131 \pm 10$ m to reach the goal. Notably, in all the experiments, irrespective of the classification model employed for navigation, the platform had to be lifted over a large fallen tree that blocked the forest trail, as there were no traversable paths to circumvent the obstacle; the incident occurred once in each replicate.

Samples of the navigation performance of the MobileNet-V1 model in different forest scenes are shown in Figure 8 (see more examples in the demonstration video of the Supplementary Material). The platform is accurately directed to perform GS actions when there are no obstacles blocking its path, despite motion blur in the input RGB image (see an example of a clear trail in dense vegetation in Figure 8A). Additionally, the classification model was able to steer the platform towards open spaces to avoid potential collisions (see Figure 8B and C—turning towards the trail in diffuse and high contrast lighting). In scenarios where the robot was facing a close-range obstacle, or large untraversable areas in the distance, the GB action was successfully triggered to avoid potential collisions (see Figure 8D—a fallen tree trunk covered in weeds and moss). Relatedly, the GB action was unnecessarily triggered only once, across all the experiments, when the platform encountered a fallen tree trunk and turned back rather than passing through the small hole between the trunk and the trail (see Figure 8E).

**Table 3.** The distance and turning rate in following a forest trail from start-to-goal in the Southampton Common woodlands for different weather conditions and times of day. Data were generated by employing the MobileNet-V1 and MobileNet-V2 models with input RGB images of $128 \times 96$ resolution. Details on lighting and weather conditions are listed in Table A6 of Appendix A.

| | MobileNet-V1 | | MobileNet-V2 | |
|---|---|---|---|---|
| Trial | Distance (m) | Turning Rate | Distance (m) | Turning Rate |
| Run 1 | 146 | 0.24 | 154 | 0.38 |
| Run 2 | 116 | 0.22 | 134 | 0.46 |
| Run 3 | 134 | 0.28 | 130 | 0.59 |
| Run 4 | 114 | 0.23 | 119 | 0.60 |
| Run 5 | 101 | 0.22 | 136 | 0.50 |
| Run 6 | 137 | 0.24 | 123 | 0.54 |
| Run 7 | 125 | 0.25 | 124 | 0.45 |
| Run 8 | 112 | 0.20 | 122 | 0.59 |
| Run 9 | 102 | 0.21 | 129 | 0.55 |
| Run 10 | 115 | 0.26 | 135 | 0.58 |

**Off-trail forest navigation:** Experiments were performed in two unfrequented areas of the Southampton Common woodlands, labeled site A and site B, spanning around 400 m$^2$ and 200 m$^2$ of forest, respectively. The two sites included obstacles such as forest litter, standing trees and fallen tree branches. The sites differed in the nature of their environment (see examples in Figure 6B). Site A had a high density of slender trees; it, however, had a very narrow corridor between waypoints for the mobile platform to slide through gaps between trees, requiring only a few turns to reach the destination. By contrast, site B comprised larger trees, tree stumps and fallen tree trunks on an uphill terrain.

Due to the small area of the off-trail environment, a round trip between waypoints was performed for each experiment. The platform was first steered by the navigation algorithm from the start to an intermediate waypoint. On reaching the intermediate waypoint, the platform was oriented back towards the start waypoint to navigate back to it. The experiment was terminated when the platformed reached the start waypoint. In our experiments, the start waypoints were located at (5056.1448 N, 124.0316 W) for site A and (5056.1568 N, 124.0155 W) for site B. Intermediate waypoints for the round trip were at (5056.1533 N, 124.0418 W) for site A and (5056.1666 N, 124.0240 W) for site B. Steering actions GS, TL, TR, GB and waypoint were used to navigate the platform. For the waypoint action, as the goal was always visible to the operator, the waypoint direction was updated through visual observation.

The off-trail experiments were performed ten times for each of the MobileNet-V1 and MobileNet-V2 classification models in several different weather conditions and times of day (see details on environmental conditions in Table A7 of Appendix A). Across all the replicates, for both the classification models, the mobile platform was able to successfully complete the round-trip path without sustaining any collisions, irrespective of the off-trail site, time of day and weather conditions. The platform steered by the MobileNet-V1 model traversed an average distance of $28 \pm 5$ m (mean $\pm$ SD across 20 replicates from both sites A and B) in the round-trip, with a turning rate of $0.13 \pm 0.08$ (see Table 4). As with the forest trail experiments, the MobileNet-V2 model was less efficient in navigation, accumulating a higher average distance of $33 \pm 7$ m to complete the round trip, and requiring a higher turning rate of $0.24 \pm 0.08$ to avoid obstacles; the turning rate was significantly higher in site B, which comprised a high density of forest vegetation (Kruskal–Wallis test, $p < 0.001$).

**Table 4.** The distance and turning rate when navigating a round trip between two waypoints off-trail in the Southampton Common woodlands. Experiments in site A and site B had start waypoints at (5056.1448 N, 124.0316 W) and (5056.1568 N, 124.0155 W), and destination way-points at (5056.1533 N, 124.0418 W) and (5056.1666 N, 124.0240 W). Data was generated by employing the MobileNet-V1 and MobileNet-V2 models with input RGB images of resolution $128 \times 96$. The details on lighting and weather conditions are listed in Table A7 of Appendix A.

| | MobileNet-V1 | | MobileNet-V2 | |
|---|---|---|---|---|
| Trial | Distance (m) | Turning Rate | Distance (m) | Turning Rate |
| Site A | | | | |
| Run 1 | 23 | 0.04 | 40 | 0.21 |
| Run 2 | 22 | 0.00 | 28 | 0.15 |
| Run 3 | 22 | 0.06 | 26 | 0.28 |
| Run 4 | 24 | 0.04 | 25 | 0.12 |
| Run 5 | 23 | 0.10 | 26 | 0.16 |
| Run 6 | 25 | 0.09 | 26 | 0.06 |
| Run 7 | 32 | 0.03 | 25 | 0.33 |
| Run 8 | 30 | 0.06 | 25 | 0.14 |
| Run 9 | 29 | 0.03 | 26 | 0.19 |
| Run 10 | 28 | 0.09 | 28 | 0.27 |
| | | | | |
| Site B | | | | |
| Run 1 | 37 | 0.21 | 38 | 0.36 |
| Run 2 | 24 | 0.20 | 45 | 0.33 |
| Run 3 | 25 | 0.17 | 36 | 0.38 |
| Run 4 | 24 | 0.17 | 41 | 0.22 |
| Run 5 | 23 | 0.18 | 39 | 0.26 |
| Run 6 | 40 | 0.14 | 41 | 0.30 |
| Run 7 | 31 | 0.22 | 37 | 0.22 |
| Run 8 | 32 | 0.20 | 38 | 0.23 |
| Run 9 | 31 | 0.20 | 37 | 0.22 |
| Run 10 | 30 | 0.31 | 37 | 0.30 |

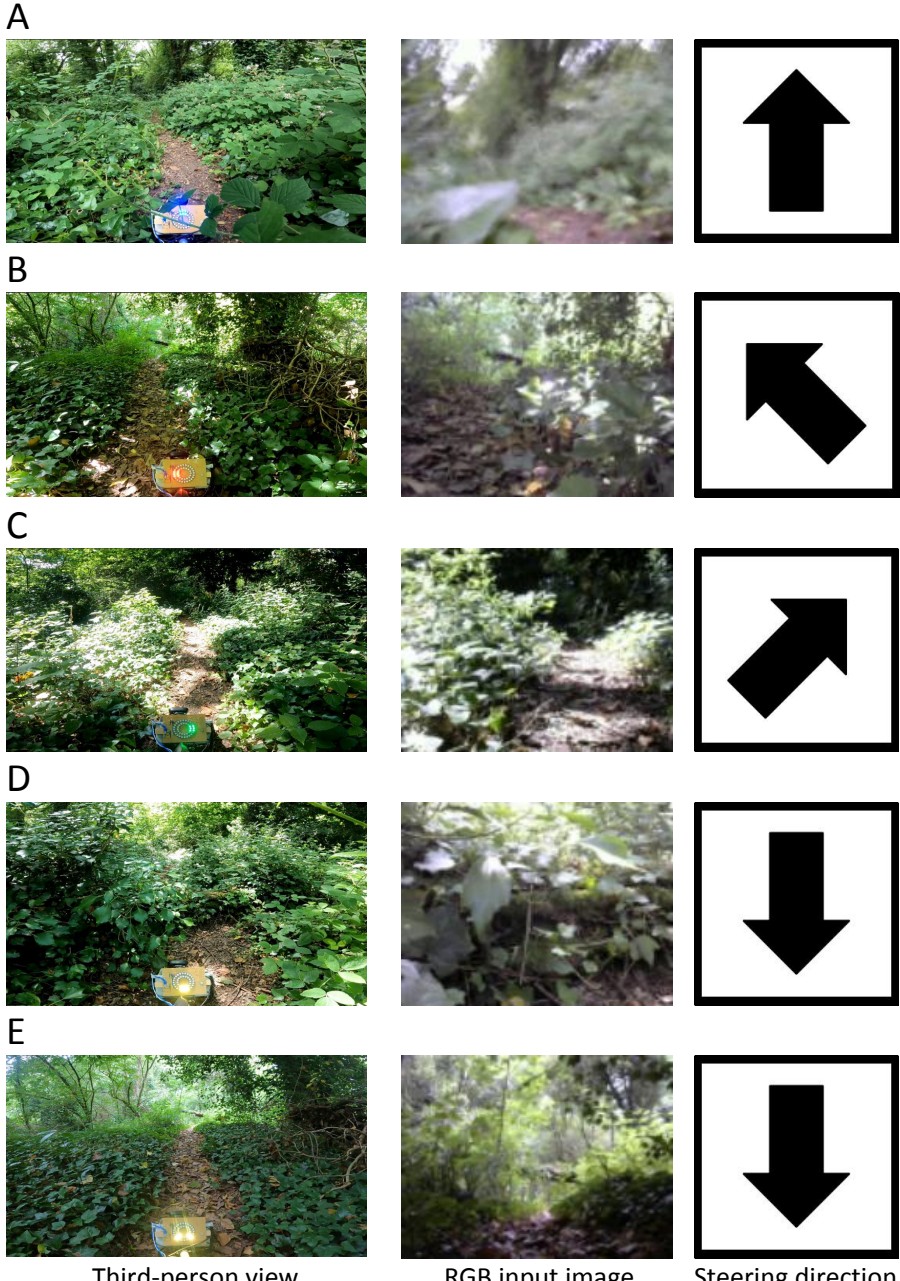

Third-person view       RGB input image       Steering direction

**Figure 8.** Steering directions output by the MobileNet-V1 model on encountering different obstacles on the forest trail at the Southampton Common woodlands, with the input RGB images at resolution 128 × 96. The corresponding 1920 × 1080 high resolution RGB images (from the GoPro camera) display a third-person view of the forest scene and the steering commands on the LED rings of the mobile platform. (**A**) blurred but clear trail across dense vegetation; (**B**) clear trail on the left of the platform, tall grass and bushes on the right and ahead of the platform; (**C**) clear trail on the right, dense bushes occupying the left area and part of the central area in front of the platform; (**D**) fallen tree trunk covered with vegetation with no clear trail in the navigation camera's field of view; and (**E**) a clear trail in front of the platform with a hanging fallen tree trunk far away from the platform that appears in the lower middle region of camera's field of view. The RGB input images displayed here have been upsampled by a factor of 10 for visual clarity.

The performance of the MobileNet-V1 model in navigating the off-trail areas of the forest is illustrated with some examples in Figure 9 (see more examples in the demonstration video of the Supplementary Material). Despite low lighting conditions, the mobile platform

was successfully steered between the narrow space among slender trees (see Figure 9A). It was able to avoid obstacles with a sequence of turning actions (see examples in Figure 9B,C of the platform avoiding a standing tree and tree stump). Moreover, to avoid potential collisions, the GB action was accurately triggered (see Figure 9D of a long fallen tree trunk). Finally, as with the forest-trail experiments, the GB action was unnecessarily triggered only once when the platform failed to identify a narrow gap between two slender trees that it could be pushed through (see Figure 9E).

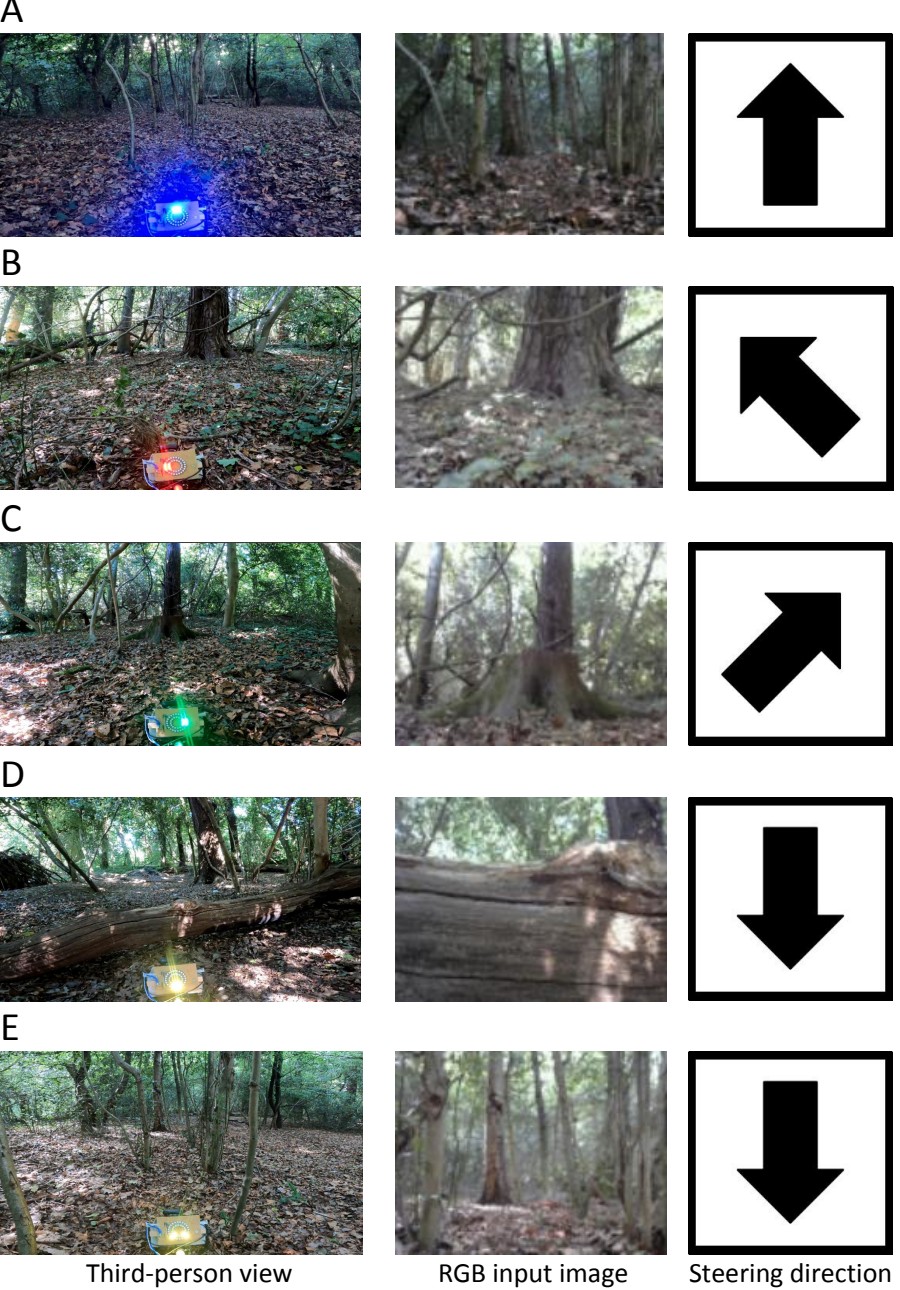

Third-person view          RGB input image          Steering direction

**Figure 9.** Steering directions predicted by MobileNet-V1 model navigated by the mobile platform on encountering different obstacles off-trail in the Southampton Common woodlands. The steering directions are annotated in the third column. (**A**) slender trees in front of the platform; (**B**) large standing trees on the right side; (**C**) tree stump and standing tree in front of the platform, tree branches and bent trees on the left side; (**D**) large fallen tree trunk in front of the platform. (**E**) slender trees in front of the platform. The RGB input images displayed here have been upsampled by a factor of 10 for visual clarity.

## 4. Discussion

In this study, we have implemented a low-viewpoint navigation algorithm for inexpensive small-sized mobile platforms navigating forest environments. For navigation, an end-to-end learning model was trained to predict steering directions from RGB images of a monocular camera mounted on the mobile platform to direct the platform towards open traversable areas of the forest, while avoiding obstacles. A multi-sensor mobile platform was used to collect training data in a forest environment, totaling almost 30,000 low-viewpoint RGB images and the corresponding rotary encoder data. We trained four state-of-the-art lightweight convolution neural networks—DenseNet-121, MobileNet-V1, MobileNet-V2 and NASNetMobile—for multiclass classification of steering actions for RGB images. From the four models, the MobileNet-V1 and MobileNet-V2 were selected for field experiments due to their high accuracy and runtime performance. Our navigation algorithms were extensively tested in real-world forests under several different weather conditions and times of day. In field experiments, using $128 \times 96$ resolution monocular RGB images, the mobile platform was able to successfully traverse a total of over 3 km of forest terrain comprising small shrubs, dense bushes, tall grass, fallen branches, fallen tree trunks, ditches, small mounds and standing trees.

The developed multiclass classification model solely relies on appearance-based information for navigation. The addition of geometry-based information may potentially provide for a better discrimination of obstacles (e.g., small close-range obstacles vs. large obstacles in the distance) with similar visual features, and consequently enable more accurate steering actions. Geometry and appearance-based information have been successfully combined in few previous studies on end-to-end learning. For example, LiDAR sensor data have been integrated with RGB images from a camera as combined inputs for navigation in indoor environments [32,36]. Our classification models may be easily extended with the addition of geometry-based information. Moreover, for low-cost platforms, depth prediction models may be employed instead of expensive depth sensors such as LiDAR (e.g., see our previous study on low-viewpoint depth prediction models for forest environments [64]).

Rovers operating in a forest are required to make safe and accurate steering decisions on a priori unknown and dynamically changing forest terrain. Therefore, a representation of the confidence of the predicted steering actions is essential for the navigation system [65]. In our classification model, the distribution of the activation of the steering output neurons may be used to approximate the uncertainty in the selected action. More principled approaches such as Gaussian process models and Bayesian deep neural networks appear promising, but computationally expensive, to infer the uncertainty in steering directions, and consequently plan safe paths for the rover (e.g., [66,67]). Finally, hardware or behavior-based solutions (e.g., see [68,69]), to nudge and probe obstacles such as grass and dense bushes, may be integrated onto the rover platform to actively reduce the uncertainty in scene understanding.

The training data for our multiclass steering classification models are captured using a mobile platform steered by an operator walking through the forest. Consequently, the operator's decisions on what obstacles may be overcome (e.g., pushing through grass, or rolling over a small fallen branch) will be distilled into the navigation algorithm of the rover. However, the training data for the steering classification models may be generalized to rovers with more advanced locomotion capabilities. Obstacles that could be overcome by a rover with better climbing ability than assumed by the operator will only occupy the area at the lower edge of the image frame. Such frames may be identified with image processing to automatically remove them from the training or relabel them with texture discrimination filters as compliant obstacles that may be successfully pushed through.

The aim of our study was to investigate the feasibility of using end-to-end learning for steering a small-sized platform at a low viewpoint through the forest. For our field experiments, coarse steering actions of turn-left and turn-right were employed for navigation. However, our approach could easily be extended to directly output wheel speeds to a rover, using techniques such as deep reinforcement learning [47,70]. Moreover, for the training of

such a rover controller, the captured RGB images could be automatically labeled with a finer resolution of velocity vectors using the rotary encoder data from our mobile platform.

## 5. Conclusions

In this study, a mobile platform running our navigation algorithm is pushed by an operator, guided by the displayed steering directions onboard the platform. Such an approach enabled us to focus solely on the challenges of forest navigation without the additional constraints of field experiments with physical rovers, not to mention the enormous challenges in designing a portable high-endurance and low-cost off-road rover. However, our approach to navigation may be employed on real rovers. For navigation, the monocular camera on our mobile platform is mounted 18 cm over the ground, consistent with the low viewpoint of off-road small-sized rovers (e.g., see rovers deployed in [21,71,72]). Moreover, our navigation algorithm is robust to blurred images from the platform's movement as well as shadows, high-contrast lighting and low-lighting conditions. Arguably, our approach to forest navigation for small-sized rovers is promising for physical validation on real rovers.

**Supplementary Materials:** A demonstration video of our field experiments, performed on a sunny day in the morning, is available at https://www.youtube.com/watch?v=UbY4i1xodx8, accessed 25 November 2022.

**Author Contributions:** Conceptualization, C.N., D.T. and K.-P.Z.; methodology, C.N., D.T. and K.-P.Z.; software, C.N.; validation, C.N.; investigation, C.N.; resources, C.N., D.T. and K.-P.Z.; data curation, C.N.; writing—original draft preparation, C.N. and D.T.; writing—review and editing, C.N., D.T. and K.-P.Z.; visualization, C.N.; supervision, D.T. and K.-P.Z.; project administration, C.N.; All authors have read and agreed to the published version of the manuscript.

**Funding:** This research received no external funding.

**Institutional Review Board Statement:** Not applicable.

**Informed Consent Statement:** Not applicable.

**Data Availability Statement:** Not applicable.

**Acknowledgments:** The authors acknowledge the use of the IRIDIS High Performance Computing Facility, and the associated support services at the University of Southampton, in the completion of this work.

**Conflicts of Interest:** The authors declare no conflict of interest.

## Appendix A

**Table A1.** A comparison of studies on steering prediction following end-to-end learning, with the weight of the rover and the approximate cost of the sensors required for navigation. Terrains are categorized in ascending order of difficulty. Sensor costs were obtained from vendor sites, where available. Dashed lines indicate the corresponding data was unavailable.

| Reference | Environments | Approximate Sensors Cost (GBP) | Weight of Rovers (kg) |
|---|---|---|---|
| | **5: Highways and traffic road** | | |
| [44] | Racing track on traffic road | 210 | 1231 |
| [40] | Highways (sunlight facing the camera, high contrast sunlight, shadows, covered in snow) | 14,000 | 1579 |
| [37] | Traffic road | 380 | - |
| [38] | Traffic road and walkways in parks | 8400 | - |
| | **4: Off-road on cemented paths, short grass, pebbles, dirt and dry leaves** | | |
| [41] | Off-road racing track | 300 | 22 |
| [42] | Mowed and short grass off-trail | 380 | 35 |
| [43] | Cemented and off-road trails with pebbles, dirt, sand, grass and fallen leaves, with few obstacles | 5500 | 35 |

**Table A1.** *Cont.*

| Reference | Environments | Approximate Sensors Cost (GBP) | Weight of Rovers (kg) |
|---|---|---|---|
| | 3: Sidewalks and walkways in urban environment | | |
| [47] | Static environments: walkways in office areas, laboratory space and corridors; dynamic environments: sidewalks among crowds. | 355 | 12 |
| [45] | Paved road cemented on grass | 800 | 50 |
| [46] | Mowed lawn, short grass, and trees in urban environment | 1000 | 17 |
| [53] | Sidewalks outside malls and office buildings | 5500 | - |
| [48] | Walkways in neighborhoods and parks | 60 | 5 |
| [39] | Parking lots, city roads and sidewalks | 60 | 62 |
| [25] | Corridor indoors and stone trail outdoors | 400 | 30 |
| | 2: Factory floor and cluttered room indoors | | |
| [49] | Factory floor | 230 | 13 |
| [34] | Corridor indoors with few obstacles | 340 | 7 |
| [35] | Corridor, kitchen and laboratory space | 600 | 2 |
| [36] | Cluttered corridor indoors | 7000 | - |
| [50] | Cluttered maze-like indoor environment | 4000 | 7 |
| [51] | Room with few obstacles | 100 | 4 |
| [52] | Corridor indoors | 400 | 55 |
| | 1: Colored track indoors | | |
| [32] | Colored track indoors with few obstacles | 320 | 2 |
| [33] | Colored track indoors | 30 | - |
| [31] | Colored tracks indoors and outdoors, and room with few obstacles | 50 | - |
| [30] | Colored track indoors | 6 | 2 |

**Table A2.** Dataset of RGB images for the multiclass classifier. The RGB images of the dataset were labeled go straight (GS), turn left (TL), turn right (TR) and go back (GB) using the wheel encoder data of the mobile platform. Subsequently, subsets of the dataset were used for training (70%), validation (15%) and testing (15%) the multiclass classifier models.

| Data | GB | GS | TL | TR | Total |
|---|---|---|---|---|---|
| Training set | 2005 | 13,697 | 2122 | 2466 | 20,290 |
| Validation set | 432 | 2939 | 458 | 533 | 4362 |
| Testing set | 431 | 2937 | 457 | 528 | 4353 |

**Table A3.** Performance of the MobileNet-V1 models trained on different input image resolutions. Accuracy and log-loss were aggregated across 4353 images (testing set).

| Input Image Resolutions | Accuracy | Log-Loss | Model Size |
|---|---|---|---|
| $32 \times 32$ | 0.33 | 1.38 | 16 MB |
| $64 \times 64$ | 0.78 | 1.00 | 22 MB |
| $128 \times 96$ | 0.96 | 0.19 | 39 MB |
| $128 \times 128$ | 0.96 | 0.22 | 47 MB |

**Table A4.** The 5-fold cross validation for 224 × 224 resolution images among the DenseNet-121, MobileNet-V1, MobileNet-V2 and NASNetMobile models.

|  | Densenet-121 | MobileNet-V1 | MobileNet-V2 | NASNetMobile |
|---|---|---|---|---|
| Accuracy | 0.978 | 0.980 | 0.970 | 0.958 |
| Log-loss | 0.116 | 0.118 | 0.154 | 0.178 |
| Precision (macro avg) | 0.974 | 0.974 | 0.964 | 0.942 |
| Precision (weighted avg) | 0.978 | 0.980 | 0.970 | 0.958 |
| Recall (macro avg) | 0.964 | 0.962 | 0.952 | 0.944 |
| Recall (weighted avg) | 0.978 | 0.980 | 0.970 | 0.958 |
| f1-score (macro avg) | 0.968 | 0.970 | 0.958 | 0.942 |
| f1-score (weighted avg) | 0.978 | 0.980 | 0.970 | 0.958 |

**Table A5.** The 5-fold cross validation for 128 × 96 resolution image between the MobileNet-V1 and MobileNet-V2 models.

|  | MobileNet-V1 | MobileNet-V2 |
|---|---|---|
| Accuracy | 0.950 | 0.896 |
| Log-loss | 0.228 | 0.448 |
| Precision (macro avg) | 0.938 | 0.862 |
| Precision (weighted avg) | 0.952 | 0.912 |
| Recall (macro avg) | 0.930 | 0.880 |
| Recall (weighted avg) | 0.950 | 0.896 |
| f1-score (macro avg) | 0.934 | 0.860 |
| f1-score (weighted avg) | 0.950 | 0.900 |

**Table A6.** The times of day and weather conditions for all of ten experiments employing the MobileNet-V1 and MobileNet-V2 models in the forest trail environment in the Southampton Common woodlands.

| | MobileNet-V1 | | MobileNet-V2 | |
|---|---|---|---|---|
| Trial | Times of Day | Weather Conditions | Times of Day | Weather Conditions |
| Run 1 | Afternoon | Cloudy | Forenoon | Scattered clouds |
| Run 2 | Afternoon | Scattered clouds | Midday | Partly sunny |
| Run 3 | Late afternoon | Part cloudy | Afternoon | Mostly clear |
| Run 4 | Late afternoon | Clear | Afternoon | Partly sunny |
| Run 5 | Late afternoon | Mostly clear | Forenoon | Partly sunny |
| Run 6 | Forenoon | Partly sunny | Midday | Sunny |
| Run 7 | Forenoon | Clear | Afternoon | Sunny |
| Run 8 | Forenoon | Scattered clouds | Afternoon | Clear |
| Run 9 | Midday | Mostly clear | Morning, | Partly sunny |
| Run 10 | Midday | Part cloudy | Afternoon | Scattered clouds |

**Table A7.** The times of day and weather conditions for all of ten experiments employing MobileNet-V1 and MobileNet-V2 models in site A and site B of off-trail environments in the Southampton Common woodlands.

| | MobileNet-V1 | | MobileNet-V2 | |
| --- | --- | --- | --- | --- |
| Trial | Times of Day | Weather Conditions | Times of Day | Weather Conditions |
| Site A | | | | |
| Run 1 | Forenoon | Sunny | Morning | Clear |
| Run 2 | Forenoon | Sunny, mostly shadow | Morning | Partly sunny |
| Run 3 | Midday | Partly sunny | Morning | Mostly clear |
| Run 4 | Midday | Partly sunny, mostly shadow | Afternoon | Mostly clear |
| Run 5 | Afternoon | Mostly clear | Afternoon | Scattered clouds |
| Run 6 | Afternoon | Clear | Near sunset | Part cloudy |
| Run 7 | Afternoon | Scattered clouds | Near sunset | Clear |
| Run 8 | Midday | Partly cloudy | Near sunset | Scattered clouds |
| Run 9 | Afternoon | Cloudy | Evening | Cloudy |
| Run 10 | Forenoon | Sunny, mostly shadow | Evening | Cloudy |
| | | | | |
| Site B | | | | |
| Run 1 | Midday | Sunny | Morning | Clear |
| Run 2 | Midday | Sunny, Sun diffuse | Forenoon | Partly sunny |
| Run 3 | Midday | Sunny | Forenoon | Sunny |
| Run 4 | Afternoon | Partly sunny, Sun diffuse | Midday | Sunny |
| Run 5 | Afternoon | Mostly clear, Sun diffuse | Noon | Bright |
| Run 6 | Forenoon | Cloudy | Noon | Sunny |
| Run 7 | Forenoon | Scattered clouds | Afternoon | Sunny |
| Run 8 | Afternoon | Clear | Afternoon | Sunny |
| Run 9 | Midday | Mostly clear | Sunset | Clear |
| Run 10 | Midday | Clear, Partly sunny | Sunset | Mostly clear |

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
