# Peer review of "End-to-End Learning for Visual Navigation of Forest Environments"

_forests, doi:10.3390/f14020268_

Round 1

Reviewer 1 Report

This paper employs existing deep learning architectures to
predict steering directions using RGB images. The paper is well-presented and experimental results demonstrate the feasibility of deep learning
in tackling the classification task pertaining to predicting steering direction.
However, the following comments need to be addressed:

-Combine training, validation, and testing, then perform
5-fold cross-validation and report  performance measures in multiclass classification such as precision,  Recall, and F1 (Macro- (and Micro-) averaging).

-Report p-values to assess the statistical difference between the models

-In Lines 165-166 at page 5, it is stated that "
models had been pre-trained on the ImageNet dataset [60]. They were
further trained on our forest training data via transfer learning"
Need to mention which layers did you (un)freeze

Reviewer 2 Report

This paper employs an end-to-end learning for visual navigation of forest environments. In general, this paper is interesting and needs the following revisions:

1.Introduction should be simplified and condensed,which should be more about your own research 

2.Figure 1 can be better visualized if arranged like figure 2 which is not over the boundary of the context .

3. If it is possible that the direction result can be more detailed like it can show the degree of the turn 

4. In line 147,”they” should be corrected as “there”

5. Too much “we” and “our” are used in the article 

6. Without the gps or other location signal,how to confirm the directions chosen is the best 

7. More detail should be explained about the process of the determination of the direction when facing the obstacles.

8. Some neural network based navigation methods can be considered in the literature, e.g., A Neural Network-Based Navigation Approach for Autonomous Mobile Robot Systems, Applied Sciences; A Robust and Efficient UAV Path Planning Approach for Tracking Agile Targets in Complex Environments, Machines; Explainable Intelligent Fault Diagnosis for Nonlinear Dynamic Systems: From Unsupervised to Supervised Learning, IEEE Transactions on Neural Networks and Learning Systems; A comparative review on mobile robot path planning: Classical or meta-heuristic methods? Annu. Rev. Control.

Round 2

Reviewer 1 Report

-In Tables A4 and A5,
The "Avg" column corresponds to five-fold cross-validation results.
Therefore, replace words "Avg" with the corresponding model
and remove columns with five performance results (i.e., X,X,X,X,X)
. For example in Table A5, there would be two columns
MobileNet-V1 and MobileNet-V2 replaced with the third
and fifth columns respectively. Do the same for Table A4  and update captions by removing "Avg...."

--For Kruskal-Wallis test, just report p-values I.e., (e.g., p<0.001)
No need for "d.f."

-Freezing layers meaning that we utilize previous knowledge.
Therefore, need to update lines 184-187
mentioning only about unfreezing any layer in studied models
and re-training them on new data

Author Response

We thank the reviewer for their comments and suggestions which we have addressed in the submitted revision. The modifications in the revised version of the manuscript are tagged and marked in red colour. We summarise below the revisions to the manuscript.

Point 1:

-In Tables A4 and A5,

The "Avg" column corresponds to five-fold cross-validation results.

Therefore, replace words "Avg" with the corresponding model

and remove columns with five performance results (i.e., X,X,X,X,X)

. For example in Table A5, there would be two columns

MobileNet-V1 and MobileNet-V2 replaced with the third

and fifth columns respectively. Do the same for Table A4  and update captions by removing "Avg...."

Response 1: We thank the reviewer for these suggestions. The tables have now been revised accordingly.

Point 2:

--For Kruskal-Wallis test, just report p-values I.e., (e.g., p<0.001)

No need for "d.f."

Response 2: we have removed "d.f." and just reported the p-values.

Point 3:

-Freezing layers meaning that we utilize previous knowledge.

Therefore, need to update lines 184-187

mentioning only about unfreezing any layer in studied models

and re-training them on new data

Response 3: We have now clarified this in the manuscript.

Reviewer 2 Report

The authors have addressed all my comments.

Author Response

Point 1:

The authors have addressed all my comments.

Response 1:

Thanks for your patience and time in reviewing the manuscript, we highly appreciate your comments and suggestions.